# Can Respiration Complexity Help the Diagnosis of Disorders of Consciousness in Rehabilitation?

**DOI:** 10.3390/diagnostics13030507

**Published:** 2023-01-30

**Authors:** Piergiuseppe Liuzzi, Antonello Grippo, Francesca Draghi, Bahia Hakiki, Claudio Macchi, Francesca Cecchi, Andrea Mannini

**Affiliations:** 1IRCCS Fondazione Don Carlo Gnocchi ONLUS, Via di Scandicci 269, 50143 Firenze, Italy; 2Istituto di BioRobotica, Scuola Superiore Sant’Anna, Viale Rinaldo Piaggio 34, 56025 Pontedera, Italy; 3Dipartimento di Medicina Sperimentale e Clinica, Universita di Firenze, Largo Brambilla 3, 50134 Firenze, Italy

**Keywords:** disorders of consciousness, autonomic nervous system, respiration rate complexity, approximate entropy, instrumental assessment of consciousness, ECG-derived respiration

## Abstract

Background: Autonomic Nervous System (ANS) activity, as cardiac, respiratory and electrodermal activity, has been shown to provide specific information on different consciousness states. Respiration rates (RRs) are considered indicators of ANS activity and breathing patterns are currently already included in the evaluation of patients in critical care. Objective: The aim of this work was to derive a proxy of autonomic functions via the RR variability and compare its diagnostic capability with known neurophysiological biomarkers of consciousness. Methods: In a cohort of sub-acute patients with brain injury during post-acute rehabilitation, polygraphy (ECG, EEG) recordings were collected. The EEG was labeled via descriptors based on American Clinical Neurophysiology Society terminology and the respiration variability was extracted by computing the Approximate Entropy (ApEN) of the ECG-derived respiration signal. Competing logistic regressions were applied to evaluate the improvement in model performances introduced by the RR ApEN. Results: Higher RR complexity was significantly associated with higher consciousness levels and improved diagnostic models’ performances in contrast to the ones built with only electroencephalographic descriptors. Conclusions: Adding a quantitative, instrumentally based complexity measure of RR variability to multimodal consciousness assessment protocols may improve diagnostic accuracy based only on electroencephalographic descriptors. Overall, this study promotes the integration of biomarkers derived from the central and the autonomous nervous system for the most comprehensive diagnosis of consciousness in a rehabilitation setting.

## 1. Introduction

Breathing is a periodic physiological activity, supplying the organism with essential energetic substrates and removing metabolism byproducts. Respiratory rates, heart rates and muscle functions are considered non-invasive parameters to evaluate the activity of the Autonomic Nervous System (ANS) [1]. Differently from the Central Nervous System (CNS), the ANS is entitled to maintain homeostatic balance without conscious control. This two-way interaction between ANS and peripheral (heart, lung, glands) oscillations has been described via an integrative system, the Central Autonomic Network (CAN) model [2]. The latter links functionally the autonomic and cognitive modulation of peripheral rhythm and functioning [2,3] via limbic structures, brainstem, cerebellum and specific pre-frontal cortex areas [3]. The complexity of this interaction has been shown to be reduced in either traumatic [4], hemorrhagic [5] or anoxic [6] severe Acquired Brain Injury (sABI) patients. In particular, a reduction in ANS–CNS complexity and responsiveness to external stimuli was previously found in patients with a prolonged Disorder of Consciousness (pDoC) [7].

pDoCs include patients with an unresponsive wakefulness syndrome (UWS) and in a minimally conscious state (MCS), differing by responses to stimuli (respectively, reflex or intentional). pDoCs may persist or evolve toward a full recovery, emerging from an MCS (EMCS) [8]. The examination and related diagnosis of consciousness in patients with a pDoC is challenging, since the neurological status is heterogeneous, behavioral responses are inconsistent, low and variable across trials and days, thus favoring misclassification [9,10,11]. Thus, as suggested by the European Guidelines [9], it is fundamental to combine clinical and instrumental evaluations in order to minimize the risk of misdiagnosis, especially for what concerns the subtle difference in consciousness levels between patients in a UWS and MCS.

Currently, the Coma Recovery Scale—Revised (CRS-R, [12]) has become the reference scale for the clinical evaluation of patients in a pDoC recommended with minor reservations for the clinical diagnosis by the American Congress of Rehabilitation Medicine [13]. 

However, the evaluation of non-reflex behaviors (highest items on each CRS-R sub-scale) is strongly affected by vigilance fluctuations [11,14], impairments in the sensory/motor networks as, for example, severe spasticity [15], associated neurological disease including critical illness polyneuropathy [16] and diffuse pain [17]. Precautions to reduce the risk of misdiagnosis have been suggested, such as the repeated administration of the CRS-R evaluation during 5 consecutive days [11,14] and the use of a mirror for the visual pursuit assessment [18].

Thus, to further reduce the misdiagnosis rate, neuroimaging and neurophysiological tools have been identified as instrumental tools to be combined with clinical consciousness assessment [9]. Within this context, a number of predictive parameters of the diagnosis and prognosis of consciousness in a rehabilitation setting have been reported, including clinical parameters [19], electroencephalography markers [20,21] and functional magnetic resonance imaging [22]. This turned out to be indispensable, given the taxonomy refinement of consciousness with the introduction of covert consciousness [23,24], expressions as cognitive motor dissociation (CMD, [25]), higher-order motor dissociation (HMD, [26]) and behavioral-neuroimaging contrasting results as in the MCS* cohort [27].

Currently, EEG is the most used sole neurophysiological assessment performed in clinical daily practice in pDoC rehabilitation, more frequently with low-density set-ups (19 channels), thus allowing for medical reporting but with limited quantitative elaborations [28]. EEG results in a non-invasive, inexpensive instrument capable of performing point-of-care assessments also in patients with physical impairments (e.g., parenteral nutrition, craniectomies, tracheostomy). In particular, a recent systematic review by Ballanti et al. (2022) highlights how most studies performing qualitative inspection of the EEG recording in pDoCs include among the markers/predictors of consciousness background frequency, presence of cortical reactivity and antero-posterior brain reorganization (antero-posterior gradient, APG). Furthermore, previous studies showed how the presence of reactivity and APG and higher frequency content are indicators of better consciousness state and more favorable recovery [20,21,29,30,31].

However, it is known how cortical modulation of peripheral sensory functions is influenced by concurrent cognition in healthy individuals and, thus, differs across mental states [32] and sleep stages [33]. Authors investigated how autonomic functions are conditioned by different consciousness states. Riganello et al. already reported evidence on the prognostic [34] and diagnostic [35] capability of Heart Rate Variability (HRV) in patients with a pDoC. Patients with higher HRV complexity were found to be likely reaching favorable outcomes and such complexity distinguished the two cohorts. Additionally, in a larger cohort, no significant differences were found in heart rate between UWS and MCS patients during resting state recordings [36]. However, the interval between an auditory stimulus and the subsequent cardiac R-peak was smaller in MCS patients than UWS, highlighting the presence of residual processing within the direct connection between the central and peripheral systems. Additionally, Heartbeat-Evoked Responses (HER, corresponding to brain responses to ascending cardiac inputs) have been shown to correlate with glucose metabolism in the default mode network in the right superior temporal sulcus and in the right ventral occipitotemporal cortex in patients with a pDoC and, thus, to levels of consciousness [37].

Coherently with these reports, since respiration rate variability (RRV) is considered an indicator of ANS activity, our hypothesis is that RRV complexity could provide useful information in the context of a multimodal assessment of consciousness in patients with a pDoC. Respiratory patterns, even if they are not related to conscious activity, are a proxy of ANS functions. Therefore, proper ANS functioning retains both diagnostic and prognostic implications in patients with consciousness alterations. Already, with the Full Outline of UnResponsiveness (FOUR, [38]) score, respiration patterns were known to characterize neurocritical patients in an acute UWS/coma. Such a scale tests brainstem reflexes and provides information on the level of brainstem injury, including four different components: eye, motor, brainstem and respiration. Within the respiratory component, respiration patterns were grouped in mechanically supported breathing, in irregular breathing, Cheyne–Stokes breathing and regular breathing. The FOUR score’s validity and predictive value are well documented in patients in ICU [38], but it does not test all the behavioral criteria to diagnose an MCS condition [39], it does not distinguish MCS+ from MCS- [39] and more investigation is needed about its usefulness in a rehabilitation setting [39,40]. 

Thus, given the importance of having proxies of autonomic functions in pDoCs and the need for introducing new correlates of consciousness capable of disentangling the clinical diagnosis from a purely behavioral-based procedure, we aimed to: (i) extract the RR complexity (Approximate Entropy, ApEN), (ii) compare it across consciousness groups and (iii) investigate whether the addition of ApEN to known EEG diagnostic markers (cortical reactivity, background frequency and APG) can improve the accuracy of a model for consciousness differential diagnosis.

## 2. Materials and Methods

### 2.1. Study Design and Data Collection

This is an ad interim analysis of a prospective observational study [41] concerning 202 non-sedated patients consecutively admitted to IRCCS Fondazione Don Carlo Gnocchi of Florence from 1 January 2021 to 1 March 2022. Inclusion criteria were diagnosis of an sABI, adults (age > 18) and time post-onset < 4 months. Exclusion criteria were presence of mechanical ventilation, presence of a sub-tentorial lesion (cerebellar, brain stem, etc.). Approval from the local Ethical Committee was obtained (N. 16606_OSS) and enrollment was performed following the Helsinki Declaration after obtaining a written consent signed by a legal guardian. Based on the maximum score among five repetitive CRS-R administrations within seven days [14], a clinical diagnosis of consciousness was formulated (UWS, MCS or EMCS) [41]. At least twenty minutes of polygraphy ECG-EEG recording at a sample rate of 128 Hz were performed using a digital machine (Gal NT, EBNeuro) and an EEG prewired 19 electrodes head cap (Fp1-Fp2-F7-F8-F3-F4-C3-C4-T3-T4-P3-P4-T5-T6-O1-O2-Fz-Cz-Pz) set according to the 10–20 International Standard System adopting previously proposed EEG recording parameters [21,29,42,43]. In particular, recordings were filtered with a low-pass filter (cut-off frequency in the 30–70 Hz), a high-pass filter (with time constant 0.1–0.3 s) adjusted according to interpretation needs (standard gain set to 7 uV/mm, sensitivity gain 2–10 uV/mm) as in Scarpino et al. [29,42]. EEG labeling was performed by the agreement of two expert neurologists according to American Clinical Neurophysiology Society (ACNS) terminology [44]. Among the EEG descriptors, we included background frequency, cortical reactivity and antero-posterior gradient (APG). With frequency, it is intended as the rate per second of the principal oscillations in the EEG background, and it is classified as delta, theta or alpha. APG is a spatial and frequential EEG background feature characterized by an anterior-to-posterior gradient of voltages and frequencies such that in anterior derivations a lower amplitude and a faster background frequency is found and in posterior derivation the opposite happens. A patient was labeled with APG if at any point in the recording there was clear and persistent EEG activity (at least 1 continuous minute). With cortical reactivity, it is intended as any change in cerebral EEG activity following stimulation. This may include change in amplitude or frequency, including EEG amplitude reduction. Appearance of eye-blink or of muscle activity does not qualify as reactivity. Patients were labeled as non-reactive if no EEG variation was depicted after either auditory, light tactile or noxious stimulation. All EEG descriptors were coded as binary categorical variables, with 1 indicating the presence of APG, reactivity and theta background, respectively, and 0 the absence of APG, reactivity or delta frequency. Tachogram was extracted using the Python library NeuroKit2 and was visually inspected for missing beats. After extraction of R-peaks, ectopic and abnormal beats were removed by interpolation. Then, respiratory rates were computed following Van Gent et al. [45] with the NeuroKit2 library. In particular, after deriving the R-peaks, the ECG instantaneous rate was computed and band-pass filtered with a second order Butterworth filter (from 0.1 to 0.4 Hz) to obtain the respiratory rates (RR). Then, the ECG-derived respiration rate (EDR) is used to compute the Approximate Entropy.

### 2.2. Approximate Entropy

Approximate Entropy adopts a non-negative number to define the complexity of a time series. In particular, given a time series $S\left(i\right)=\left\{s\left(1\right),\;s\left(2\right),\;\ldots ,s\left(N\right)\right\},$ take m consecutive points to form $S\left(i\right)=\left\{s\left(i\right),\;s\left(i+1\right),\;\ldots \;s\left(i+m-1\right)\right\}$ and define the distance between $S\left(i\right),\;S\left(j\right)$ as follows
$$d\left[S\left(i\right),S\left(j\right)\right]=max_{k=0,1,\ldots ,m-1}\;\left|s\left(i+k\right)-s\left(j+k\right)\right|$$

Then, for a given threshold r, evaluate the number of distances smaller than r divided by the total number of distances (*N* – *m* + 1) recorded as $C_{m}^{r}\left(i\right)$.

After taking the natural logarithm of $C_{m}^{r}\left(i\right)$ and calculating the average across all N points, the quantity $\phi_{m}\left(r\right)$ can be defined. Then the dimension of the m coefficient was increased to m+1 and the Approximate Entropy was computed as follows:$$ApEn\left(m,r,N\right)=\phi_{m}\left(r\right)-\phi_{m+1}\left(r\right)$$

Signal elaboration was performed on a workstation with two Intel Xeon Silver 4216 16 Cores each, 256 GB RAM KSM26RD4 and a 32 GB GeForce RTX3090 GPU.

### 2.3. Statistical Analysis

Outcome was set as a three-class categorical variable (UWS, MCS, EMCS). Numerical variables (ApEN) entered a Kruskal–Wallis test for multiple group comparison targeting the consciousness state. Conditioned to its significance, Dunn–Bonferroni post hoc tests were conducted to cope with multiple comparisons. Similarly, chi-square analyses were conducted for EEG binary descriptors (background frequency, reactivity and anteroposterior gradient) with *z*-tests for multiple comparisons (with the Bonferroni correction). ApEN also entered a univariate ordinal logistic regression and a multivariate one together with the included EEG descriptors. Such descriptors also entered a multivariate ordinal regression without the presence of ApEN in order to compare results. Then, the Wilks test was adopted as likelihood-ratio test to ensure that the addition of each predictor significantly decreased the −2log likelihood (−2LL) of the full model in contrast to the ones of the reduced models (i.e., the full model without the specific predictors). All analyses were performed on SPSS (Vs 26, Chicago, IL, USA, SPSS Inc.) and a *p*-value < 0.05 was considered as statistically significant.

## 3. Results

Two hundred and forty-nine patients were enrolled in the study, of which 34 were excluded (31 patients with mechanical ventilation, 29 suffering from a sub-tentorial lesion and 18 excluded for both reasons). Therefore, 207 patients were included in the analysis. Of these 207, 5 were excluded due to missing data, resulting in 202 patients with a median age of 64 years [IQR = 22], 42 UWS, 61 MCS, 99 ECMS, with an overall median CRS-R score of 15 points [IQR = 12] (Table 1A). Of the 72 patients with a traumatic injury, 19 patients had a right lesion, 22 a left one, 14 a bilateral one, 4 a sub-tentorial one and 13 suffered from diffuse axonal injury (DAI). The median execution time to extract ApEN for one patient was of 1.41 s [IQR = 0.23 s] (2 × Xeon Silver 4216 3.2 Ghz, 256 GB Ram, RTX 3090 24 GB).

Significant differences (*p* = 0.001) were found by a Kruskal–Wallis test between ApEN and the three consciousness groups (UWS, MCS and EMCS) (Table 1B). In particular, via Dunn–Bonferroni post hoc tests, significant differences were detected between UWS and MCS (*p* = 0.007, UWS _mean rank_ = 70.77 and MCS _mean rank_ = 106.29) and between UWS and EMCS (*p* < 0.001, EMCS _mean rank_ = 111.59), also visible from Figure 1. Additionally, conditioned to chi-square test significance, the presence of theta background frequency and APG differed between EMCS and MCS and EMCS and UWS (*p* < 0.05). On the other hand, cortical reactivity was found to be significantly different between all pairs of consciousness states, increasing in higher levels of consciousness.

Comparing ApEN across etiologies (Kruskal–Wallis) resulted in a significant difference across groups (χ(3) = 13.108, *p* = 0.003), with the only pairwise comparison (Dunn post hoc test) remaining significant after the Bonferroni correction being the one between traumatic (median ApEN 1.12 [IQR = 0.27]) and hemorrhagic (median ApEN 1.03 [IQR = 0.29]) patients (p^FDR^ = 0.002).

Then, given the significant differences in distributions between etiologies and consciousness states (chi-square, χ = 8.896, *p* = 0.012), specifically between UWS and EMCS (Bonferroni-corrected multiple proportions *z*-tests, *p* < 0.05), we investigated whether there is a significant interaction effect of etiology and consciousness on the ApEN. A Generalized Linear Model (GLM) was applied with ApEN as response variable and etiology and consciousness as fixed factors (Table 2) with related Dunn–Bonferroni post hoc tests across both fixed factors.

ApEN was found to be significantly different across consciousness levels (F = 10.731, *p* < 0.001) and across etiologies (F = 3.014, *p* = 0.031) (Table 2). However, no significant interaction effect was found between consciousness and etiology (F = 2.136, *p* = 0.051). Post hoc analysis within etiologies showed how hemorrhagic patients have significantly lower ApEN than traumatic ones (*p* = 0.001, p_FDR_ = 0.004) and how EMCS patients have higher ApEN than either MCS (p_FDR_ = 0.003) or UWS (p_FDR_ = 3 × 10^−5^). The same analysis was repeated, grouping etiology as TBI/non TBI, with similar results. Hence, ApEN was found to be significantly different across consciousness levels (F = 6.932, *p* = 0.001) and across TBI/non-TBI (F = 6.126, *p* = 0.014). Similarly, the interaction consciousness *x* TBI was found to be not significant (F = 0.364, *p* = 0.695) and the post hoc analysis (within consciousness levels) lead to the same results of the previous analysis.

Ordinal logistic regression analysis with the target set to the consciousness diagnosis showed a decreased RRV in patients with either a UWS or MCS (*p* = 0.002, β = 1.994, 95%CI = 0.818–3.568, Table 3A). When evaluating diagnostic capability of EEG descriptors, a positive effect was found to be impressed by the presence of theta background frequency (compared to delta) and the presence of cortical reactivity (*p* = 0.001 and *p* < 0.001, respectively, Table 2B). Coherently, when EEG descriptors and ApEN entered a multivariate analysis (Table 2C), higher RR complexity (ApEN) was found to be related to better consciousness levels (MCS, EMCS, *p* = 0.002) as well as theta frequency and cortical reactivity. Nagelkerke R^2^ increased from 0.321 to 0.393 when ApEN was added to the set of independent variables. Model C resulted in a −2LL of 338.108 with all reduced models resulting in a higher −2LL. In particular, a −2LL of 361.584 (*p* = 8 × 10^−6^), of 349.321 (*p* = 0.004), of 353.571 (*p* = 4 × 10^−4^) and of 350.189 (*p* = 0.002) was found for the reduced ApEN, frequency, APG and reactivity models, respectively.

Of the entire cohort, only 12.4% (N = 25) had seizures during their acute stay. Additionally, respectively, 17 and 4 patients had epileptic seizures and paroxysmal sympathetic hyperactivity during the rehabilitation stay (Table 4). On the other hand, the majority of patients with epileptic seizures were treated with non-benzodiazepine-based drugs.

## 4. Discussion

In this work, we provided evidence of how RR complexity differs between different states of consciousness and how the combination of EEG-based parameters and information on the RR complexity significantly improved the model performances in contrast to solely EEG-based data. Such evidence is important, since the assessment of consciousness levels in patients with a pDoC remains a challenging task. Identifying subtle and hardly detectable signs of consciousness is crucial, since they are related to a better prognosis [19]. Multifactorial approaches, including instrumental evaluations based on clinical, EEG, ECG and neuroimaging biomarkers have been proposed and are supported by international guidelines [9]; however, these evaluations do not allow a clear distinction between UWS and MCS. With this work, we seek to open a new line of investigation on the quantitative assessment of respiration patterns, and their variability, for the assessment of the lowest states of consciousness (UWS, MCS). Evidence of correlation between ANS functioning and consciousness alterations are well documented [7,37]. In neurocritical care, in ICU, the evaluation of breathing patterns in unresponsive patients is already routinely performed, seeking diagnosis of specific respiration patterns (e.g., apnea, Cheyne–Stokes) and to confirm a diagnosis of likely brain stem lesions. Such assessment was included in the FOUR score, validated in neurocritical patients. Furthermore, being two of the five possible scores on the respiratory sub-item of the FOUR score connected to mechanically ventilated patients and two connected to two different types of brainstem-related respiratory patterns, given the exclusion criteria of this study, we can assert that the analyzed cohort is much less critical than the one to which the FOUR score applies. However, even if possibly supported by instrumental monitoring, the FOUR score is strictly related to a clinical assessment of the respiration pattern. On the other hand, to our knowledge, no previous study investigated the association between quantitatively derived EDR and consciousness levels in patients in the rehabilitation phase. The present work aimed to introduce, beside the recommended tools for the diagnosis of patients in a pDoC, an innovative evaluation of respiration based on instrumental assessment. Coherently with our hypothesis, we found that higher complexity of respiratory rates is indeed associated with higher levels of consciousness and improves the diagnostic capability of EEG descriptors alone. To this extent, our results are in line with literature findings concerning variability and cardiac responsiveness [8], confirming that a better CNS–ANS interaction is a proxy of higher states of consciousness. Additionally, variability of the derived EDR complexity may be a useful proxy of pain perception. In particular, breathing responses are discarded in nociception clinical scales because of the difficulty in reliably visually assessing breathing patterns in non-mechanically ventilated patients (who are frequently found in sub-acute and chronic settings) [46]. Thus, EDR complexity, which is assessed automatically and quantitatively, may be indeed used as a proxy of pain perception as already performed with HRV-based [47] indicators in pDoCs.

We acknowledge that respiratory waveforms derived from ECG are less precise than photoplethysmography; however, they are clearly sufficient to detect respiratory rates and to derive their complexity. Indeed, it has been shown how ECG is sufficient to detect respiration peaks both when the patient is still and when he/she is moving [48,49]. The algorithm used in this work was validated with a mean error of 0.028 Hz (median of 0.015 Hz, maximum of 0.102 Hz) from the PPG-based reference estimate [45]. Such algorithms, even if they are not validated on patients with a brain injury, had earlier validation both in surgery studies and during different motions (e.g., running, driving [49,50,51,52,53]), thus ensuring good performances even with small unvoluntary movements and hyperidrosis (common in pDoCs). Additionally, before computation of RRV, the ECG signal was low-passed with a Butterworth filter with a cut-off frequency of 5 Hz, as suggested in Soni et al. [54] and in the NeuroKit2 tool [55]. Then, focusing on the 0.1–0.5 Hz frequency band of the FFT transform of the ECG recording, algorithms are used to detect the respiration peak only within that window, thus ensuring almost no HRV-based contamination noise affects the RRV. Furthermore, reducing the complexity of the experimental set-up and relying only on instruments used in the clinical routine fosters the translational potential of this solution to clinical practice. Deriving respiration from single-use ECG electrodes would also reduce the presence of redundant devices and, thus, the burden on caregivers.

Whether respiration complexity can be considered a correlate of consciousness in patients with brainstem lesions is an issue deserving a separate analysis. Due to the small number of patients with sub-tentorial lesions in our sample (N = 29), we removed them from our analysis. Therefore, this issue needs to be tackled in further, larger studies. Another limitation of our study is the possible influence of paroxysmal sympathetic hyperactivity and consequent benzodiazepine-based treatments on heart and respiratory rate, although already very limited in our population (Table 3). Additionally, the influence of critical respiratory complications (e.g., pulmonary infections, hyperpyrexia) on RR complexity has to be tackled in further analyses, since these data were not collected for all patients and it is known from previous studies that it may affect the outcome of patients with a pDoC [56,57,58]. Lastly, it has been shown how disagreements between brain metabolism and behavioral diagnosis of consciousness are mostly related to patients with borderline CRS-R scores (i.e., UWS patients with a CRS-R of 6 or 7) [59]. In such patients, the diagnosis made on the best CRS-R sub-scores is not always coherent with the PET-based one. In these cases, some of the UWS patients may exhibit brain metabolism typical of the presence of consciousness (non-behavioral MCS, MCS*) but, when assessed with the CRS-R, only reflex activity is detected. We must acknowledge that this calls for a complementary PET-based diagnosis of consciousness and that the inclusion of such assessment would be a further outlook of the study. However, the CRS-R being the current reference scale for the assessment of patients with a pDoC, such a limitation has always to be taken into account in the absence of PET-based diagnosis.

## 5. Conclusions

Diagnosis of patients with a pDoC is complex. By exiting from a strictly neuro-centric approach, this work provides evidence that an inexpensive and non-invasive index of respiration complexity, a proxy of central autonomic network functioning, allows for a better discrimination between different levels of consciousness compared to EEG biomarkers alone. The latter, as already conducted with quantitative analysis of HRV, suggests that direct assessment of respiration rate complexity can be a relevant indicator to be added to the multimodal consciousness assessment protocol to distinguish UWS and MCS patients.

## Figures and Tables

**Figure 1 diagnostics-13-00507-f001:**
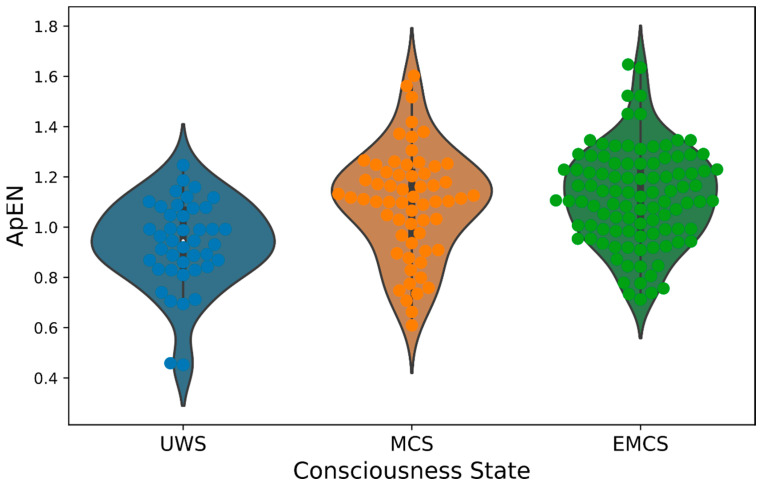
Approximate Entropy (ApEN) of respiration rate (RR) intervals across consciousness states. **Legend.** ApEN: Approximate Entropy; UWS: Unresponsive Wakefulness Syndrome; MCS: Minimally Consciousness State; EMCS: Emergence from MCS.

**Table 1 diagnostics-13-00507-t001:** Descriptive statistics for the entire cohort and the individual subgroups for demographic variables (**A**) and instrumental variables (**B**). Median and interquartile ranges (in brackets) are reported for numerical independent variables whilst count and percentages (in parentheses) are reported for categorical ones. *p*-values are reported for chi-square analysis for categorical variables (EEG descriptors) and Kruskal–Wallis test for numerical ones (ApEN). Two columns sharing a subscript (^+^, ^†^, ^‡^) are not significantly different (after Bonferroni correction) after the post hoc pairwise analysis.

	Demographic Variables				
A	Total sABI (N = 202)	UWS (N = 42)	MCS (N = 61)	EMCS (N = 99)	
Age, *years*	64 [22]	67 [18]	66 [24]	62 [24]	
Gender, *M*	113 (55.9%)	26 (61.9%)	31 (50.8%)	56 (56.6%)	
Etiology					
*Traumatic*	72 (35.6%)	8 (19.0%)	20 (32.8%)	44 (44.4%)	
*Anoxic*	13 (6.4%)	7 (16.7%)	1 (1.6%)	5 (5.1%)	
*Ischemic*	34 (16.8%)	6 (14.3%)	9 (14.8%)	19 (19.2%)	
*Hemorrhagic*	83 (41.1%)	21 (50.0%)	31 (50.8%)	31 (31.3%)	
TPO, *days*	47.5 [25]	49.50 [31]	47 [25]	45 [26]	
CRS-R, *points*	15 [12]	7 [8]	15 [9]	23 [3]	
**B**	Polygraphy variables				Significance
Frequency, *Theta*	67 (33.1%)	4 (9.5%) ^+^	14 (23.0%) ^+^	49 (49.5%) ^‡^	*p* < 0.001
APG, *Present*	63 (31.2%)	8 (19.1%) ^+^	16 (26.2%) ^+^	39 (39.4%) ^‡^	*p* = 0.025
Reactivity, *Present*	85 (42.1%)	9 (21.4%) ^+^	24 (39.3%) ^‡^	52 (52.5%) ^†^	*p* = 0.002
ApEN, *#*	1.08 [0.29]	0.97 [0.22] ^+^	1.11 [0.34] ^‡^	1.10 [0.28] ^‡^	*p* < 0.001

**Legend.** M: Males; TPO: Time Post-Onset; CRS-R: Coma Recovery Scale—Revised; APG: Anteroposterior Gradient; ApEN: Approximate Entropy; UWS: Unresponsive Wakefulness State; MCS: Minimally Conscious State; EMCS: Emergence from MCS; #: pure number, no measurement unit.

**Table 2 diagnostics-13-00507-t002:** Generalized linear model with ApEN as response variable.

	F	*p*-Value
*Consciousness State*	10.731	**<0.001**
*Etiology*	3.014	**0.031**
*Consciousness x Etiology*	2.136	0.051

**Legend.** Variables in bold are found to be significant.

**Table 3 diagnostics-13-00507-t003:** Ordinal regression results for only ApEN (**A**), EEG descriptors (**B**) and a combination of the two (**C**).

	Ordinal Logistic Regression			
	*p*-Value	β	95%C.I	
**A**: R^2^ = 0.081				
ApEN	0.002	1.994	0.818	3.568
**B**: R^2^ = 0.321				
Frequency, Theta	0.001	1.077	0.417	1.738
APG, present	0.171	0.526	−0.227	1.279
Reactivity, present	<0.001	2.243	1.407	3.080
**C**: R^2^ = 0.393				
ApEN	0.002	2.239	0.846	3.632
Frequency, Theta	0.001	1.136	0.463	1.808
APG, present	0.131	0.586	−0.174	1.346
Reactivity, present	<0.001	2.165	1.328	3.002

Legend. ApEN: Approximate Entropy; APG: Antero-Posterior Gradient. Reference is set to going toward EMCS.

**Table 4 diagnostics-13-00507-t004:** Descriptive statistics for epileptic (during ICU and IRU) and neurovegetative crisis (during IRU). Specific numerosity is provided for patients intaking benzodiazepine-based drugs (BDZ). One patient may take more than one anti-epileptic medication. With paroxysmal sympathetic hyperactivity (PSH), it is intended as referring to a state of excessive sympathetic nervous system activity characterized by sudden increases in heart rate, blood pressure, respiratory rate, temperature, dystonia, rigidity and spasticity.

	Total sABI (N = 202)
Epileptic crisis during ICU stay	25 (12.4)
Epileptic crisis during IRU stay	17 (8.4)
*Clonazepam*	1
*Diazepam*	1
*Piracetam*	0
*Midazolam*	0
*Non-BDZ*	17
PSH during IRU stay	4 (2.0)
*Fentanyl*	3
*Diazepam*	0
*Non-BDZ*	3

Legend. ICU: Intensive Care Unit; IRU: Intensive Rehabilitation Unit; BDZ: benzodiazepine; PSH: paroxysmal sympathetic hyperactivity.

## Data Availability

The data that support the findings of this study will be made available from the corresponding author upon request for replication/research purposes.

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
