# Peer review of "Can Respiration Complexity Help the Diagnosis of Disorders of Consciousness in Rehabilitation?"

_diagnostics, 2023, doi:10.3390/diagnostics13030507_

Round 1

Reviewer 1 Report

This paper describes a method to evaluate the functioning of the ANS through the variability of the respiratory rate in patients with pDOC.

The manuscript is overall well written and raises important issues in the evaluation of these patients, providing a tool that is potentially usable at bedside and could benefit from a translation to clinics.

I have a few minor points to address and one major issue for which I would like to hear the arguments of the authors of this paper.

*Minor

The introduction could benefit from some references about ANS function in pDOC and especially on HRV (for instance, papers of Riganello et al.).

*Major (?)

The paper relies entirely on the extraction of the data based on the ECG. However, the ECG is subject to its own variability, i.e. heart-rate variability (HRV). HRV has been reportedly suggested to be modified according to the level of consciousness. Therefore, deriving the respiratory rate data based on the the variability of the ECG signal is a potential methodological flaw that could have major consequences in the presented results... Indeed, RRV is of course influenced by the HRV if extracted from the ECG signal... + the algorithm used was not validated in a brain injured population as I think it was done in a population of patients undergoing surgery and it is diffuclt to extrapolate that the observation are the same in this population as in patients with DoC..

For this reason I would like that the author argue why they choose this path and that the consequences are at least discussed in the paper.

All in all, with the reserve related to my comments (and response to them), I would recommend the paper for publication.

Author Response

A point-by-point reply is provided within the attached .docx file.

Reviewer 2 Report

This study examined the effect of respiration rate complexity in distinguishing between the different persistent disorders of consciousness (pDoC) states. The authors conducted a prospective study in 202 pDoC patients and recorded ECG and EEG for at least 20min. They then computed approximate entropy from breath rate derived using the ECG signal, and compared that with EEG metrics in the form of anterior-posterior gradient, reactivity, and background frequency for differentiating different pDOC groups of UWS, MCS, and E-MCS. They found that respiration rate approximate entropy improved logistic regression model performance in differentiating the patient groups compared to using EEG metrics alone.   This study had an interesting premise to examine complexity of respiration rate as an additional marker in assessing consciousness state in pDOC. However, there are numerous issues with both methodology and presentation of the manuscript that will need to be addressed before this manuscript can be considered for publication.    Major Concerns   Introduction:  -The authors claim to investigate whether RR complexity could be a proxy of prompt responses to environmental changes. However, the introduction does not provide adequate background information and citations to justify the connection between RR complexity and consciousness, and linking that with environmental changes. Additionally, the experimental design did not include any environmental manipulations, so it is unclear how the study will evaluate response to environmental changes. The introduction should better explain and justify the study goals and experimental plan.   -The introduction currently does not have sufficient background information regarding the EEG measures that are used in the study. The majority of the introduction is focused on ANS and respiration rate, but EEG measures are mentioned near the end of the introduction without appropriate context or background literature (lines 79-82). The authors should better justify the experimental approach of the current study by providing more information about the EEG metrics and relevance for consciousness evaluation, as well as including relevant citations.    Methods: -Methods section is highly inadequate. There is insufficient information about how the EEG descriptors are derived, as the raw EEG signal is recorded as sampled time series data, but the methods section only states that EEG was labeled as delta or theta, cortical reactivity, and antero-posterior gradient without providing any information about how these metrics were derived from the continuous time series data. What were the signal processing steps? Were there any notch or bandpass filtering applied? Were the EEG data properly denoised prior to extraction of the metrics? Additionally, are the frequency-based metrics based on quantitative EEG (qEEG) approach derived from the entire 20min of EEG data, or was the continuous data segmented into shorter epochs prior to feature extraction?    -Methods require more citations to justify experimental approaches, such as the definition and relevance of the anterior-posterior gradient and cortical reactivity for consciousness evaluation in section 2.1. The authors should also better explain how the EEG cortical reactivity was computed for this study.   -How was the antero-posterior gradient measured? Were the voltage amplitudes computed as mean over some segment of time-domain signal? Was the frequency information determined using some frequency transform such as Fourier, or a time-frequency transform such as wavelet? Was it based on some visual inspection by experts? Furthermore, what was the criterion used to determine how much difference between the anterior and posterior regions was sufficient to create the gradient (e.g. a 1% , 5%, or 10% difference in amplitude)?   -The statistical analysis section indicates that Chi-Square test was used to compare the EEG-based metrics (section 2.3, Lines 137-138). The Chi-Square test is for categorical variables, which I do not believe is appropriate in this case. The authors do not describe how they derived the EEG metrics like frequency and reactivity, but if these were computed like standard quantitative measurements, then the results are numerical values which should not be compared using the Chi-Squared test. Instead, it would be more appropriate to use an ANOVA or Kruskal-Wallis test if the EEG metrics were quantitative measurements that produce numerical values. Please clarify the exact procedure employed, and update statistical methods accordingly.    -Experimental design and data collection. The manuscript text seems to indicate that the duration of EEG-ECG recording is “at least 20min”, which suggests that the data collection duration can vary across participants. This creates inconsistent experimental parameters across the study sample, and should be properly addressed in the analysis by defining a set duration of data to be used for all subjects, and trimming away excess data where applicable. This has not been done by the authors. The manuscript does not provide any information about data segmentation employed in this study.   Results: -The authors report that the R2 value increased when they incorporated ApEN to the regression model. However, the authors should also report whether the increase in R2 is statistically significant in order to have more convincing results. Rather than using separate models, it would be a better approach to perform a hierarchical regression, such that they can demonstrate whether the incorporation of ApEN variable into the model significantly improves the amount of variance explained. Additionally, the authors should also examine potential collinearity in the regressor variables using a stepwise regression.    Discussion -The authors claim that the use of ApEN improves the diagnosis of DoC compared to EEG metrics alone (lines 191-193). This claim is not supported by the evidence presented, as the authors have not shown “diagnosis” of DoC is improved by using ApEN. Instead, the authors have reported increased R2 value in the regression model from 0.32 to 0.38 when they incorporate ApEN into the model, but they still need to examine whether this increase is statistically significance. If the increase is significant, then the authors may claim that the model performance is improved by incorporating the ApEN variable, but should not claim "diagnosis" is improved.    -Discussion: The authors claim that their measurement of respiratory rate complexity is a proxy of rapid response to environmental changes (lines 217-219), but their study has not included any environmental manipulations or measured any environmental changes. This claim is thus inappropriate and should be removed.   -Conclusion: The authors claim that their study provides evidence that an "index of respiration complexity might be treated as a proxy of connectivity changes in the autonomic network”, but their study has not measured any connectivity changes, nor provided any evidence to justify such a claim. This statement should be removed.   Minor Concerns: -Table 1: notation is confusing in part B. All 4 rows are labeled with p<0.05, with two of the columns showing ‘*’ which typically used to used to denote statistical significance. However, the table heading  confusingly states that “two columns sharing a subscript are not significantly different”. Does this mean that the p value is from the omnibus Kruskal-Wallis test? If so, which pairs are significantly different in post-hoc multiple comparisons? If the columns with * are not significantly different from one another, is the assumption then that all other columns are pairwise different after Bonferroni correction? This result needs to be clarified.   -Clarification needed: Section 2.3, what does it mean that “Outcome was declined…”? (Line 134)   -Acronyms are used in the manuscript text without definition. For example, what is ACNS? (Line 98)   -Article abstract is inadequate; Objective statement and large portions of Methods are both more appropriate for Background. This should be revised.    -Line 155: The statement "The median execution time for each patient was of 1.41 155 s [IQR = 0.23 s].” What is execution time? Please clarify/rephrase.

Author Response

(The authors gave the same response as above.)

Reviewer 3 Report

The article form Liuzzi et al concern a very interesting topic, i.e. the role of autonomic dysfunction after severe brain injuries and its prognostic value.

I have however some comments for the authors:

1.       The patients enrolled in the study are all in a chronic phase . Why so much importance is given to FOUR score ? lines 60 to 71 and 200 to 212. Four-score assess basically brainstem function in comatose acute patients, and has nothing to do with chronic DOC patients were the dysautomic dysfunction is not related to brainstem …

2.       The use of CRS-R is for sure a standard, but it has at least 20% misclassification , therefore all the assumptions based on CRS-R might be not trues. This is a limit of the study that should be discussed.

3.       Table 1 is not clear. Measurement unit of TPO should be specified . Days ? Percentages should be specified (% of traumatic cases in UWS 8/42 is much less than in EMCS 44/99…)

4.       Is there any difference in ApEN between traumatic, anoxic or vascular cases ? these data be provided

5.       Some information should be given about the brain-lesion pattern in the cohort (especially for traumatic cases). Focal lesions ? Cortical lesions ? DAI ? Anoxic lesions ?

Author Response

(The authors gave the same response as above.)

Round 2

Reviewer 2 Report

I would like to thank the authors for revising their manuscript. Although many of my comments have been addressed, there are still two remaining issues which are described below:

  The Chow test is used to determine whether the coefficients of linear regression are significantly different. However, the authors have used logistic regression modeling in their analysis, which is a classification technique and fundamentally different from linear regression. The use of the Chow test to determine statistical significance between the different models in this study is therefore incorrect. The authors should revise their analysis to more adequately assess statistical significance in their models.   The abstract still contains background information in the Objective section, and the aim statement in Methods should be more appropriately placed in the Objective section. The authors should update the abstract accordingly.  

Author Response

The point-by-point reply has been inserted in the attached file. We thank the reviewer for his/her feedback and constructive criticisms.

Reviewer 3 Report

The article is definitely improved and more clear.

One relevant point still deserves to be better clarified before publication.

ApEN is significally different between traumatic and hemorrhagic patients (line 234; values are not specified);

ApEN is significally different across different consciousness states, being maximal between UWS and EMCS (and this is the more intersting finding of the study);

however the distribution of traumatic cases is different (basically inverted) in UWS and EMCS, respectively 19% vs 44% as well as hemorrhagic cases 50% vs 31%;

is there any effect of this asymmetrical distribution of different aetiologies on the  difference of ApEN among different consciousness states ? 

Author Response

(The authors gave the same response as above.)

Round 3

Reviewer 2 Report

I would like to thank the authors for updating their manuscript. I believe the revised version is now ready for publication.